# Brassinosteroid Accelerates Wound Healing of Potato Tubers by Activation of Reactive Oxygen Metabolism and Phenylpropanoid Metabolism

**DOI:** 10.3390/foods11070906

**Published:** 2022-03-22

**Authors:** Ye Han, Ruirui Yang, Xuejiao Zhang, Qihui Wang, Bin Wang, Xiaoyuan Zheng, Yongcai Li, Dov Prusky, Yang Bi

**Affiliations:** 1College of Food Science and Engineering, Gansu Agricultural University, Lanzhou 730070, China; hanyest1989@163.com (Y.H.); y1248598953@163.com (R.Y.); zhangxuejiao8@126.com (X.Z.); wqh1030018113@163.com (Q.W.); wangbin_1519@163.com (B.W.); xiaoyuanzheng1@163.com (X.Z.); lyc@gsau.edu.cn (Y.L.); 2Department of Postharvest Science of Fresh Produce, Agricultural Research Organization, Rishon LeZion 7505101, Israel; dovprusk@volcani.agri.gov.il

**Keywords:** wound healing, potato tubers, phenylpropanoid metabolism, ROS, brassinosteroid, abiotic stress

## Abstract

Wound healing could effectively reduce the decay rate of potato tubers after harvest, but it took a long time to form typical and complete healing structures. Brassinosteroid (BR), as a sterol hormone, is important for enhancing plant resistance to abiotic and biotic stresses. However, it has not been reported that if BR affects wound healing of potato tubers. In the present study, we observed that BR played a positive role in the accumulation of lignin and suberin polyphenolic (SPP) at the wounds, and effectively reduced the weight loss and disease index of potato tubers (cv. Atlantic) during healing. At the end of healing, the weight loss and disease index of BR group was 30.8% and 23.1% lower than the control, respectively. Furthermore, BR activated the expression of *StPAL*, *St4CL*, *StCAD* genes and related enzyme activities in phenylpropanoid metabolism, and promoted the synthesis of lignin precursors and phenolic acids at the wound site, mainly by inducing the synthesis of caffeic acid, sinapic acid and cinnamyl alcohol. Meanwhile, the expression of *StNOX* was induced and the production of O^2−^ and H_2_O_2_ was promoted, which mediated oxidative crosslinking of above phenolic acids and lignin precursors to form SPP and lignin. In addition, the expression level of *StPOD* was partially increased. In contrast, the inhibitor brassinazole inhibited phenylpropanoid metabolism and reactive oxygen metabolism, and demonstrated the function of BR hormone in healing in reverse. Taken together, the activation of reactive oxygen metabolism and phenylpropanoid metabolism by BR could accelerate the wound healing of potato tubers.

## 1. Introduction

Potato tuber (*Solanum tuberosum* L.) is prone to mechanical injury in the stage of harvesting and postharvest handling [1]. The wounds on surface provided channels for the invasion of various pathogens, resulting in a large amount of tuber decay during storage. The rate of potato tuber loss after harvest is as high as 20–25%, among which the loss caused by rot accounts for 60–70% [2]. Interestingly, potato tubers could form a typical healing structure at the wound site, which effectively reduce the water loss and prevent the infection of pathogens [3,4]. However, it usually takes a long time to form typical and complete healing structures [5]. Hence, it is essential to explore methods of accelerating wound healing.

Brassinosteroid (BR), the sixth plant hormone, is a kind of phytosterol, which widely exists in plants [6]. BR is important for plant development and growth [7]. In the study of leaf inclination in rice, the phenotype of dwarf mutant *d2* could be restored to normal shape by exogenous BR [8]. BR played an important role in the etiolation process of *Arabidopsis thaliana* seedlings. The ability of the BR-deficient mutant seedlings to turn green was significantly lower than that of the wild type, which was related to the effect of BR on the key enzymes of chlorophyll synthesis [9]. BR also associates with plant resistance to both abiotic and biotic stresses [10]. BR spraying enhanced resistance of cucumber root to *Fusarium* wilt [11]. BR treatment improved the resistance of strawberry plant against *Colletotrichum acutatum* [12]. BR improved the defense response of rose petals against *Botrytis cinerea* [13]. BR was an important regulator of growth-immunity trade-off, which involved in several signal elements, such as *BRI1* and *BAK1*. Receptors *BRI1* and *BAK1* in the BR signaling pathway played an active role in innate immune signaling, and the phosphate site of *BAK1* was critical for the immune response [10]. Further studies indicated BR could increase multiple plants resistance to pathogens through promoting the synthesis of ethylene, abscisic acid (ABA) and jasmonate acid (JA, [14,15]. *BES/BZR* is one of the crucial transcription factors participating in BR signal pathway. *BZR1* was reported to interact with *WRKY40* to regulate the expression of immune-related genes in *Arabidopsis thaliana* [16]. In addition, BR improved defense resistance of *Nicotiana benthamiana* against viral diseases by promoting the synthesis of H_2_O_2_ and nitric oxide (NO), and also activating the antioxidant system [17,18]. BR promoted the amassing of phenolic acids and lignin by activating phenylpropanoid metabolism [19].

BR was able to improve plant abiotic stress resistance, including mechanical damage stress [20]. Drought, salt, low and high temperature stresses were important factors leading to abiotic stress in plants. They could lead to changes in plant morphology, physiology, biochemical metabolism and cell structure by destroying enzyme systems [8]. BR regulated the expression of target genes through a series of phosphorylation cascades with a variety of transcription factors, thus improving abiotic stress resistance in plants. For example, BR ameliorated drought and salt stresses in tomato by regulating MAP-kinase and *YODA* transcription factor to affect stomatal conductance. The application of BR was an effective way to alleviate the cold stress of plants, and can significantly increase the activities of SOD and POD [9]. BR can also alleviate the photoinhibition effect by increasing photochemical quenching coefficient, quantum number and photochemical efficiency of photosystem II. Exogenous application of BR significantly increased the photosynthetic activity of many plants under low temperature and weak light stress [21]. In addition, a large number of studies have shown that BR can crosstalk with ABA, ethylene, cytokinin and other plant hormones to jointly regulate various abiotic stresses. In wheat, the application of BR decreased the activity of cytokinin oxidase and the expression of encoding genes, thus promoting the level of cytokinin [22]. As a specific inhibitor of BR, brassinazole (BRZ) inhibits the key hydroxylation process of BR synthesis through preventing the activity of cytochrome P450 [23]. Differing from BR, the accumulation of flavonoids in tea was reduced by BRZ treatment via inhibiting the synthesis of NO [24]. Similarly, BRZ retarded grape fruit mature and anthocyanin accumulation by inhibiting the synthesis of endogenous BR and ABA [25]. BR has been widely known to enhance the resistance of plants to abiotic and biotic stresses. Whether BR affects the wound healing of potato tubers through reactive oxygen metabolism and phenylpropanoid metabolism, has not been reported.

In the present study, BR was thought to accelerate the wound healing of potato tubers by activating reactive oxygen metabolism and phenylpropanoid metabolism. The wounded potato tubers (cv. Atlantic) were treated with BR and BRZ, then the tubers were placed in the dark (20–25 °C, 85% RH) for wound healing. The aims of this study were (1) to determine the disease index and the weight loss of tubers during healing, (2) to observe the deposition of lignin and suberin at wounds, (3) to determine the expression levels of *StDWF* and *StBES*, which related to BR biosynthesis and signal pathway, (4) to analyze the genes expression and enzyme activities of PAL, CAD and 4CL, and the content of phenolic acids and lignin precursors, (5) and to analyze the expression levels of *StNOX* and *StPOD*, and the contents of O^2−^ and H_2_O_2_.

## 2. Material and Methods

### 2.1. Plant Material and Treatment

Potato tubers (*Solanum tuberosum* L. cv. Atlantic) were obtained from Ailan Potato Company in Dingxi City, Gansu Province, China. The tubers with uniform size, without diseases or mechanical damage were selected for the experiment.

After the surface washed and disinfected, the potato tubers were cut in half to simulate mechanical damage and randomly divided into three experimental groups. Each group contained 180 tubers and three biological replicates were performed. The first and second groups were treated with 10 μM 24-epicastasterone (No. S18014, Yuanye Biological Technology Co., LTD, Shanghai, China, BR group) and 40 μM brassinazole (No. S87623, Yuanye Biological Technology Co., LTD, Shanghai, China, BRZ group) for 3 min, respectively [25,26]. The treatment concentration of 24-epicastasterone and brassinazole were screened by preliminary experiment (data not shown). In contrast, the third group (CK) was treated with distilled water for 3 min as the control. Subsequently, the tubers were placed in the dark (20–25 °C, 85% RH) for wound healing.

The samples with a thickness of 2 mm at wounds were collected at 0, 1, 3, 5, 7 and 14 d after healing. After freezing with liquid nitrogen, the samples were ground into powder, and storied at −80 °C.

### 2.2. Evaluation of Wound Healing Efficacy

The weight loss was measured by gravimetric method. Each treatment contained 45 tubers and three biological replicates were performed.

The disease index was evaluated according to Yang, et al. [27]. Simply, the spore suspension of *Fusarium. sulphureum* (40 μL, 1 × 10^6^ spores/mL) was applied to the wound site of tubers at 0, 3, 5, 7 and 14 d. After 7 days, the disease index of tubers was evaluated according to the following formula:Disease index = Σ (the number of diseased wounds × the level of disease)/4 × total wounds × 100%

The level of disease was divided into 4 grades: the diseased area covering 1/4 of the wound area was classified as grade 1, and the diseased area covering 1/2 and 3/4 was classified as grade 2 and grade 3, respectively. When diseased area covering the total wound area was classified as grade 4, and no diseased area was classified as grade 0. Each treatment contained 30 tubers per time point and three biological replicates were performed.

### 2.3. Microscopic Observation of Suberin Polyphenolic (SPP) and Lignin

SPP and lignin was observed as described by Yang, Han, Han, Ackah, Li, Bi, Yang and Prusky [27]. Briefly, the tuber slices with 0.2 mm were cut on the vertical wound surface. After rinsing the starch particles, SPP was observed by autofluorescence using fluorescence microscope (BX51, Olympus Co., LTD, Tokyo, Japan). The lignin was observed with an optical microscope after staining with phloroglucinol.

The thickness cell layer of SPP and lignin in wound tissues was determined with IS Capture software according to Yang, Han, Han, Ackah, Li, Bi, Yang and Prusky [27].

### 2.4. Expression Analysis by QRT-PCR

The total RNA of potato tubers was isolated by Total RNA Kit (No. DP419, TianGen Biotech, Beijing, China), then the cDNA was gained by Fast King RT Kit (No. KR116, TianGen Biotech, Beijing, China). QRT-PCR was analyzed by Light Cycler 96 (Roche, Basel, Switzerland) and the amplification efficiencies were tested using the standard curves. The gene relative expression levels were calculated by the comparative C_T_ (2^–ΔΔCT^) method [28]. Using transcriptome data, we previously screened the key genes *StPAL*, *St4CL*, *StCAD*, *StNOX* and *StPOD* induced by wound healing (results not shown). The primers used are shown in Table 1 and *StEF1α* was served as internal control.

### 2.5. Measurement of Enzyme Activities

PAL activity was determined based on the description of Koukol and Conn [29]. Briefly, samples were homogenized in borate buffer (40 g/L PVP, 5 mM β-mercaptoethanol, 2 mM EDTA, pH 8.8) to extract enzyme. L-phenylalanine (20 mM) was used as substrate and the mixture was reacted at 37 °C for 60 min, then the absorbance at 290 nm was tested with a spectrophotometer (UV-1800, Shimadzu, Kyoto, Japan). One unit (U) of PAL activity was defined as 0.01 increase of absorbance per min.

A modified method was used to measure 4CL activity according to Li, et al. [30]. 0.6 g samples were homogenized in 3 mL Tris-HCl buffer (15 mM β-mercaptoethanol, 5 mM EDTA, 0.15% PVP and 30% glycerin, pH 8.0) to extract enzyme. Then, 450 μL MgCl_2_ (15 mM), 150 μL p-coumaric acid (5 mM), 150 μL CoA-SH (1 mM) and 150 μL ATP (50 mM) were added into 500 μL supernatant, respectively. The absorbance was measured at 333 nm after reaction at 40 °C for 30 min. One U of 4CL activity was defined as 0.01 increase of absorbance per hour.

A modified method was used to measure CAD activity according to Goffner, et al. [31]. 3.0 g samples were homogenized in 3 mL phosphoric acid buffer (15 mM β-mercaptoethanol, 2% polyethylene glycol, 0.1% PVPP, pH 6.25) to extract the enzyme. Then, 800 μL reaction solution (5 mM trans-cinnamic acid and 10 mM nicotinamide adenine dinucleotide phosphate) were added into 200 μL supernatant. The absorbance at 340 nm was measured after reaction at 37 °C for 30 min. One U of CAD activity was defined as 0.001 increase of absorbance per min.

### 2.6. Measurement of the Contents of Phenolic Acids and Lignin Precursors

The contents of phenolic acids and lignin precursors were measured according to Ayaz, et al. [32]. One gram sample was added 70% methanol (3 mL), and ultrasonicated at 40 Hz for 30 min. After freeze dried, the supernatant was redissolved in 1 mL mixture containing methanol, water and glacial acetic acid (70:30:1). Ultra-High Performance Liquid Chromatography (Waters ACQUITY Arc Bi, Milford, CT, USA) was used with Waters Symmetry C18 (4.6 mm × 250 mm, 5μm). The flow rate was 800 μL/min and 5 μL sample was added. The maximum absorbances of coniferyl alcohol, p-coumaric acid, cinnamic acid, and ferulic acid was 263 nm, 310 nm, 276 nm, and 322 nm, respectively. The maximum absorbance of sinapic acid and caffeic acid was 325 nm, while sinapyl alcohol and cinnamyl alcohol was 273 nm. The contents of lignin precursors and phenolic acids were expressed as μg/kg.

### 2.7. Measurement of the Contents of O_2_^−^ and H_2_O_2_

The content of O_2_^−^ was measured by O_2_^−^ Content Test Kit (No. A052, Jian Cheng Bioengineering Institute, Nanjing, China) following the protocol of manufacturer. The content of H_2_O_2_ was calculated by Tissue H_2_O_2_ assay Kit (No. A064, Jian Cheng Bioengineering Institute, Nanjing, China) according to Yang, Han, Han, Ackah, Li, Bi, Yang and Prusky [27]. Briefly, 0.5 g samples were homogenized in 4.5 mL normal saline to extract the H_2_O_2_, then the test was carried out following the protocol of manufacturer. The contents of O_2_^−^ and H_2_O_2_ were expressed as mol/kg.

### 2.8. Statistical Analysis

Three biological replicates were performed and the result was exhibited as mean ± standard error. The statistical significance analysis was evaluated by an ANOVA, and the means were compared by Fisher’s LSD test using SPSS Statistics 22.0. A significance level of *p* < 0.05 was used throughout the study.

## 3. Results

### 3.1. Effects of BR and BRZ Treatments on Wound Healing Efficacy

Weight loss and disease index are key indicators to assess the efficacy of wound healing [5]. The weight loss of control and treated tubers gradually increased during the healing process (Figure 1A). BR group was always lower than that of CK, however BRZ group was higher. At the end of healing (14 d), the weight loss of BR group was 30.8% lower than the control. However, the weight loss of BRZ group was 51.9% higher than the control. The disease index of the control and treated tubers decreased gradually with the extension of healing time (Figure 1B). After 5 d of healing, BR group was lower than that of CK. BRZ group, however, was higher than that of CK. Notablely, the disease index of BR group was 23.1% lower than that of CK on 14 d. However, the disease index of BRZ group was 14.8% higher than the control. The results showed that BR treatment accelerated the tubers healing, however BRZ treatment was the opposite.

During healing, the wounded surfaces layer of potato tubers became thicker, while that of the three groups were similar on 0–7 d of healing. The surfaces layer of the control was slightly more obvious than that of BRZ group on 14 d of healing (Figure 1C).

### 3.2. Effects of BR and BRZ Treatments on the Deposition of Lignin and SPP at Wounds

Lignin and SPP are the main components of healing closing layer [33]. The accumulation amount of SPP at wounds gradually increased during healing (Figure 2A,C). BR group was always higher than that in CK. However, the thickness cell layer of SPP in BRZ group was lower than that in the control on 7–14 d. Similarly, a gradual increase of lignin was found among the control and treated tubers during wound healing (Figure 2B,D). Meanwhile, BR group was also always higher than CK. The thickness cell layer of lignin in BRZ group was 20.0% lower than control on 7 d. The results shown that BR promoted the accumulation of lignin and SPP at wounds, however BRZ inhibited their deposition.

### 3.3. Effects of BR and BRZ Treatments on the Expression of StDWF and StBES

*StDWF* was a key gene involved in brassinosteroid biosynthesis, while *StBES* was one of downstream transcription factors in brassinosteroid signal pathway [34]. The gene expression of *StDWF* and *StBES* was dramatically induced by wound healing, which on 1 d was more than 10-fold in contrast with that on 0 d (Figure 3). BR treatment promoted the expression of *StDWF* and *StBES*, however BRZ treatment showed the inhibition. On 1 d of healing, the expression of *StDWF* and *StBES* in BRZ group was only 26% and 15% of the control, respectively. These results suggested that the synthesis and signal pathway of endogenous BR was induced during wound healing, and exogenous BR treatment promoted this process, however BRZ treatment was the opposite.

### 3.4. Effects of BR and BRZ Treatments on Genes Expression and Activities of PAL, 4CL and CAD

Phenylpropanoid metabolism took an essential part in tubers healing as catalyzing the formation of phenolic acids and lignin precursors [35]. The expression levels of *StPAL*, *St4CL*, *StCAD* and activities of PAL, 4CL, CAD were induced by wound healing (Figure 4). The peak of *StPAL* expression and PAL activity was exhibited on 1 d and 3 d, respectively (Figure 4A,B). The expression of *StPAL* and activity of PAL in BR group were increased rapidly during healing, while that in BRZ group increased slowly. The PAL activity in BR group was consistently higher, as seen with a 1.5-fold level compared to the control on 3 d.

The gene expression of *St4CL* reached the peak on 1 d, which was 152.4-fold in contrast with that on 0 d (Figure 4C). The maximal activity of 4CL enzyme was found on 5 d, with 3.7-fold in contrast with that on 0 d (Figure 4D). BR treatment induced the expression of *St4CL* (1 d, 5 d) and always increased 4CL activity. On the contrary, BRZ treatment inhibited 4CL activity, as seen with a 34.1% decrease compared to the control on 1 d.

Compared with PAL or 4CL, both the gene expression and the activity of CAD reached the peak slightly later (Figure 4E,F). In detail, the expression level of *StCAD* reached the peak on 3 d, while the CAD activity of three groups increased on 1 d, then maintained at a high level. BR treatment induced the expression of *StCAD* (1 d, 3 d) and increased CAD activity except on 3 d. However, BRZ treatment reduced CAD activity, with 34.1% lower level compared with the control on 7 d. These revealed that BR increased the expression levels of *StPAL*, *St4CL*, *StCAD* in tubers, and further activated PAL and 4CL at the early and middle stages of healing, and enhanced CAD activity at middle and late stages of healing. In contrast, BRZ treatment showed the inhibition.

### 3.5. Effects of BR and BRZ Treatments on the Contents of Five Phenolic Acids and Three Lignin Precursors

Caffeic acid, *p*-coumaric acid, ferulic acid, cinnamic acid, and sinapic acid are the main components of SPP, while sinapyl alcohol, coniferyl alcohol, and cinnamyl alcohol are the precursors of lignin [3]. During healing, the content of phenolic acids gradually increased, especially caffeic acid, sinapic acid and *p*-coumaric acid, which on 7 d were 1.6, 1.7 and 1.6-fold in contrast with that on 0 d, respectively (Figure 5A–E). BR treatment caused an obvious increase in contents of caffeic acid and sinapic acid, with 24.5% and 21.0% higher than CK on 7 d, respectively. On the contrary, BRZ treatment greatly inhibited the accumulation of sinapic acid, which was only 59% of the control on 7 d. During healing, cinnamyl alcohol and coniferyl alcohol were induced, which on 7 d were 72.5% and 43.6% higher than uninjured tubers, respectively (Figure 5F–H). Furthermore, BR treatment promoted the accumulation of cinnamyl alcohol, which was 1.3 times that of CK on 5 d. BRZ treatment, however, inhibited the accumulation of coniferyl alcohol on 3 d and 7 d. In addition, both BR and BRZ treatments had little effect on the content of sinapyl alcohol. In general, BR could promote the accumulation of phenolic acids and lignin precursors, mainly by inducing the synthesis of caffeic acid, sinapic acid and cinnamyl alcohol.

### 3.6. Effects of BR and BRZ Treatments on Gene Expression of StNOX and StPOD, and the Content of O_2_^−^ and H_2_O_2_

NOX could catalyze the generation of O_2_^−^ by transferring electrons to O_2_, then superoxide dismutase (SOD) catalyzed the dismutation of O_2_^−^ into H_2_O_2_ [36]. On 1 d after wound, the expression levels of *StNOX* and *StPOD* were induced by healing, which was 4.9 and 8.3-fold in contrast with that on 0 d, respectively (Figure 6A). The gene expression of *StNOX* in BRZ group was suppressed except on 7 d, while BR group was higher than the control on 1 d and 3 d. In the control, the expression of *StPOD* kept a high level at the middle stage of healing, which on 3 d was 14.4-fold that of 0 d (Figure 6B). However, BR treatment activated the expression of *StPOD* only on 1 d and 5 d, while BRZ treatment slightly inhibited the expression of *StPOD* on 3 d and 7 d.

O_2_^−^ and H_2_O_2_ are the major reactive oxygen species (ROS) participated in tubers wound healing [36]. The H_2_O_2_ content of tubers increased dramatically on 1d, and then gradually increased during healing (Figure 6C). Furthermore, consistently higher contents of H_2_O_2_ were observed in BR group, which was 51% higher than CK on 1 d. BRZ group shown lower H_2_O_2_ content, particularly in the late stage of healing. Similarly, the accumulation of O_2_^−^ could be activated by wound, but it changed irregularly during healing (Figure 6D). BR treatment accelerated the production of O_2_^−^ in comparison with the control, but BRZ treatment didn’t always inhibit its content. These results indicated that BR treatment could induce the production of O_2_^−^ and H_2_O_2_, mainly due to activate the expression of *StNOX* during tubers healing. In contrast, BRZ treatment did the opposite.

## 4. Discussion

BR is important for inducing the resistance to abiotic and biotic stresses in plants [10]. BR application in potato has been reported to improve tubers formation [37], and has also been reported to lead to tuber germination [38]. Exogenous BR could enhance endogenous BR synthesis and signal pathway by activating BR receptor *BRI* [39], BR synthesis gene *DWF* and downstream signaling transcription factor *BES/BZR* [40]. In potato, *StDWF* has been proved to be a key gene involved in the synthesis of BR, and *StBES* was a transcription factor participated in BR signal pathway [34]. During healing, the expression of *StDWF* and *StBES* genes were observably induced in comparison with that in uninjured tubers (Figure 3). Compared with the control, exogenous BR treatment activated *StDWF* and *StBES*, while BRZ treatment did the opposite. In *Arabidopsis thaliana*, exogenous BR treatment can induce the expression of *AtDWF4* and play a positive role in root elongation [41]. In grape, exogenous BR could activate the synthesis of endogenous BR through enhancing the expression of BR synthesis gene *VvDWF1*, and inducing BR receptor *VvBRI1* [39]. Meanwhile, exogenous BR treatment promoted the expression of transcription factor *TaBES2*, and thus enhanced the antioxidant capacity of wheat [42]. Therefore, we infer that exogenous BR could activate the expression of *StDWF* and *StBES* and enhanced the synthesis and signal pathway of endogenous BR, which induced tuber wound healing.

Phenylpropanoid metabolism was crucial in wound healing, which provided phenolic acids and lignin precursors for the polymerization of lignin and SPP [43]. PAL participated in the beginning of phenylpropanoid metabolism as a rate-limiting enzyme, catalyzing the conversion of phenylalanine into cinnamic acid [44]. Using cinnamic acid as substrates, a series of phenolic acids were produced through multi-step reactions, such as coumaric acid, ferulic acid, caffeic acid and sinapic acid [3]. 4CL catalyzed the synthesis of ferulic acid-CoA, coumaric acid-CoA and caffeic acid-CoA with phenolic acids as substrates [5,27]. These phenolic acids and their derivatives were assembled and polymerized to form SPP, which was deposited in the cell wall near the wounds [43]. CAD catalyzed the final step of lignin synthesis by converting aldehydes into alcohols [45]. At last, these alcohols were transported to the secondary cell wall as lignin monomers, then oxidized and crosslinked by POD to form lignin [27]. In the present study, the expression levels of *StPAL*, *St4CL*, *StCAD* and the enzyme activities corresponding to related genes could be markedly induced by BR treatment (Figure 4). By further testing the product, the contents of sinapic acid, caffeic acid, cinnamyl alcohol and coniferyl alcohol were increased in BR group (Figure 5). Similar reports indicated that BR could enhance the expression of *PAL* and *4CL* in tomato under stress, thereby increasing the accumulation of phenolic acids and flavonoids [46]. Exogenous BR could strengthen the disease resistance in tea plants by inducing expression levels of *Cs4CL*, *CsPAL* and *CsC4H* in phenylpropanoid pathway [26]. Tomato treated with BR shown higher PAL activity and higher total phenolic and proline contents, which could enhance the resistance to chilling injury [47]. Notably, the lignin accumulation of BR-mediated potentially took part in improving the resistance of garlic to salt stress [48]. In maize, BR could activate the promoters of stress-related genes through the downstream transcription factor *ZmBES1/BZR1-5*, thus promoting the thickening of secondary wall and positively improving plant resistance to salt or drought stress [6,49]. Meanwhile, endogenous BR can also promote the synthesis of secondary metabolites such as cotton fiber [50], corpus callosum [14], by affecting other hormones ethylene [51], ABA [52], or signaling molecules NO and H_2_O_2_ [24]. The effect of NO and ABA in accelerating the wound healing of vegetables and fruit through promoting phenylpropanoid metabolism has been confirmed [53,54]. Hence, we speculate that BR could activate phenylpropanoid metabolism directly by activating the involved key genes or indirectly by inducing ABA, NO or H_2_O_2_, thereby promoting the accumulation of phenolic acids and lignin precursors. On the contrary, BRZ treatment weakened the activation of phenylpropanoid metabolism by inhibiting the endogenous BR synthesis and signal pathway, or the production of ABA, NO or H_2_O_2_, thus resulting in a decrease in the accumulation of phenolic acids and lignin precursors.

The healing structure was mainly composed of suberin and lignin, of which suberin was divided into suberin polyaliphatic (SPA) and SPP [33]. SPP and lignin was mainly composed of phenolic compounds, while SPA was mainly composed of fatty acid derivatives [55]. ROS are indispensable for oxidative crosslinking of SPP and lignin, which is also important for the injury signal pathway [36]. Once damaged, the tuber quickly occurs oxygen burst at the wound site, which could produce mostly O_2_^−^ and H_2_O_2_ [56]. During healing, NOX, POD and polyamine oxidase were involved in the production of H_2_O_2_. Among these, NOX played a major role in H_2_O_2_ production [57]. Tuber injury stress could activate NOX, with catalyzing O_2_ to form O_2_^−^ [36]. Due to a short life of O_2_^−^, it would be quickly catalyzed by the disproportionation of SOD to produce H_2_O_2_, which had a long life [58]. POD was not only critical to the production of H_2_O_2_ and O_2_^−^, but also participated in the oxidizes cross-linking of lignin and SPP [59]. In comparison with the control, the accumulation of lignin and SPP in BR group was more remarkable (Figure 2). Meanwhile, BR treatment induced the expression of *StNOX*, and the contents of O_2_^−^ and H_2_O_2_ increased correspondingly (Figure 6). However, BR treatment could not always promote the expression level of *StPOD*. In tomato, BR was participating in the regulation of pollen development, and the mutation of *BZR1* could lead to the decline of ROS in tapetal cells, thus affecting pollen development [60]. Furthermore, *BZR1* could activate the promoter of *RBOH1* directly, thus affecting NOX activity and ROS production [60]. In addition, BR could effectively improve the tolerance of tomato to high temperature stress and regulate the accumulation of H_2_O_2_ by increasing the enzyme activity and gene expression of NOX [8]. Hence, we infer that the transcription factor of BR hormone signaling downstream could directly activate NOX to promote the accumulation of O_2_^–^ in tubers. Then, more H_2_O_2_ was further generated by SOD disproportionation, which was conducive to the transmission of wound stress signal and oxidative crosslinking of SPP and lignin. Both of them could prevent the invasion of pathogens and reduce the loss of water from the wound site. Moreover, BR treatment promoted the synthesis of antibacterial substances like flavonoids, which may be another factor in the decline of tuber disease index.

## 5. Conclusions

Exogenous BR treatment could promote the synthesis and signal pathway of endogenous BR by activating *StDWF* and *StBES*. BR activated the expression of *StPAL*, *St4CL*, *StCAD* genes and related enzymes activities, and promoted the synthesis of coniferyl alcohol, cinnamyl alcohol, sinapic acid and caffeic acid at the wound sites. In addition, BR promoted the activity of NOX and the production of O^2−^ and H_2_O_2_, which mediated oxidative crosslinking of above phenolic acids and lignin precursors. As more SPP and lignin was accumulated at wounds, resulting in lower disease index and less weight loss of healing tubers. The inhibitor BRZ demonstrated the function of BR hormone in healing in reverse.

## Figures and Tables

**Figure 1 foods-11-00906-f001:**
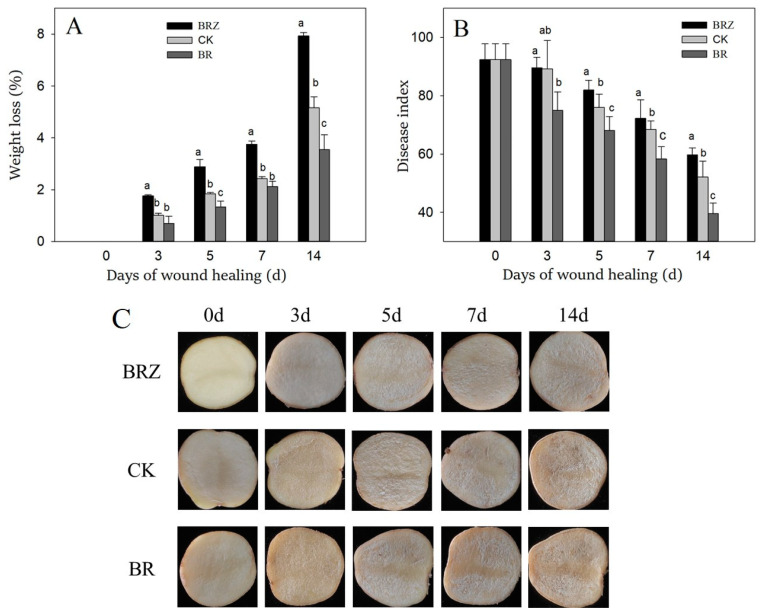
Effects of BR and BRZ treatments on weight loss (**A**), disease index (**B**) and wound appearance (**C**) of potato tubers during healing. Vertical bars indicate the standard error of three replicate assays. Columns with different letters at each time point are significantly different (LSD, *p* < 0.05).

**Figure 2 foods-11-00906-f002:**
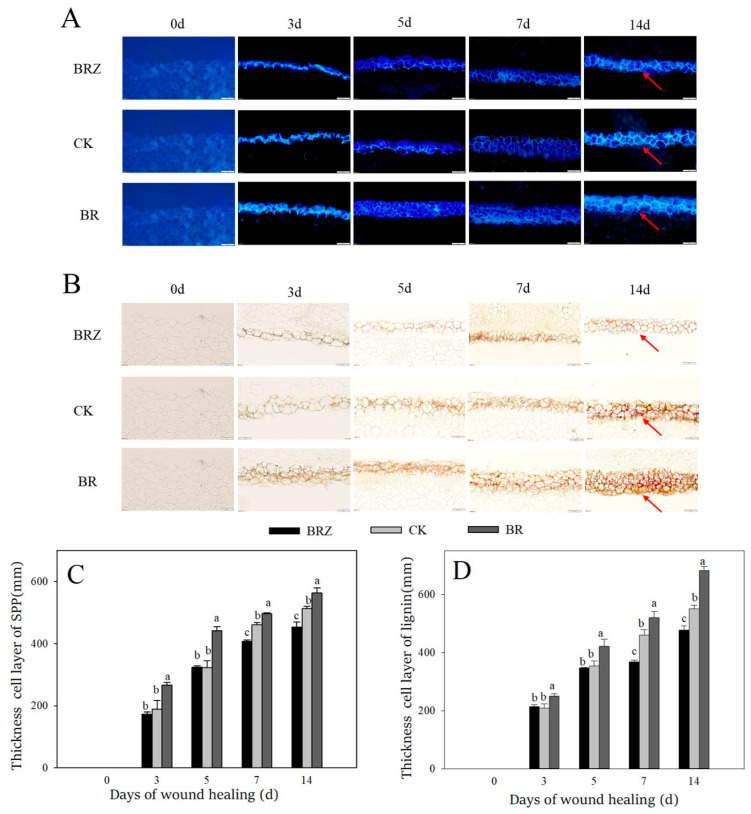
Effects of BR and BRZ treatments on the accumulation and thickness cell layer of SPP and lignin during healing. (**A**) Accumulation of SPP. (**B**) Accumulation of lignin. (**C**) Thickness cell layer of SPP. (**D**) Thickness cell layer of lignin. Arrows represent the accumulation site of SPP and lignin. The size of the scale bars is 200 μm. Vertical bars indicate the standard error of three replicate assays. Columns with different letters at each time point are significantly different (LSD, *p* < 0.05).

**Figure 3 foods-11-00906-f003:**
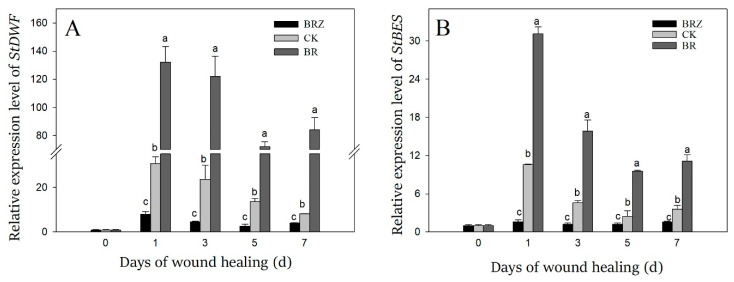
Effects of BR and BRZ treatments on the expression levels of *StDWF* (**A**) and *StBES* (**B**) genes. Vertical bars indicate the standard error of three replicate assays. Columns with different letters at each time point are significantly different (LSD, *p* < 0.05).

**Figure 4 foods-11-00906-f004:**
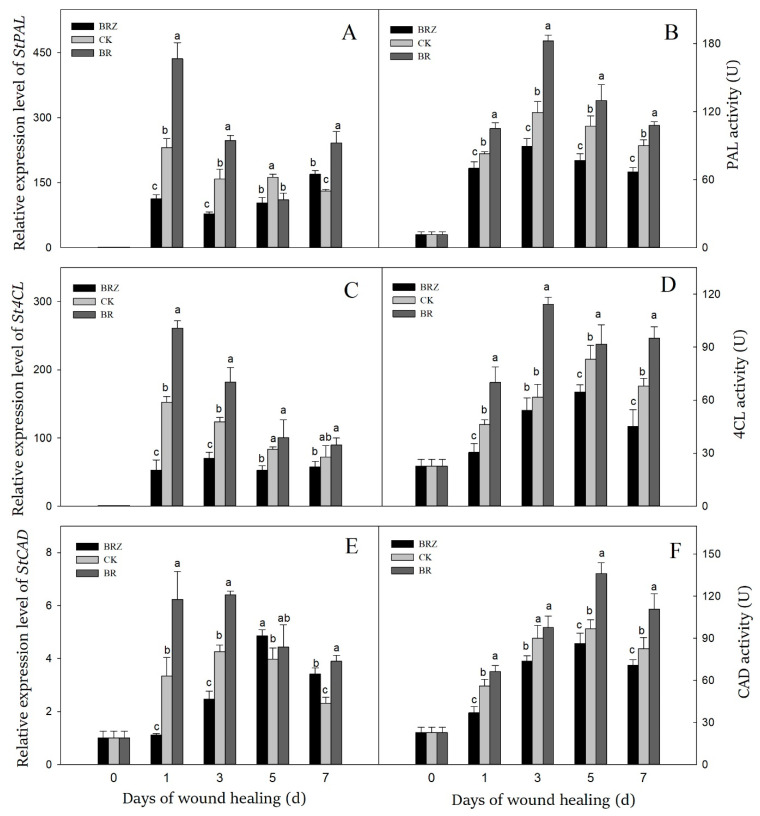
Effects of BR and BRZ treatments on the expression of key genes and related enzyme activities of phenylpropane metabolism. The expression levels of *StPAL* (**A**), *St4CL* (**C**) and *StCAD* (**E**) genes. The activity of PAL (**B**), 4CL (**D**) and CAD (**F**) enzymes. Vertical bars indicate the standard error of three replicate assays. Columns with different letters at each time point are significantly different (LSD, *p* < 0.05).

**Figure 5 foods-11-00906-f005:**
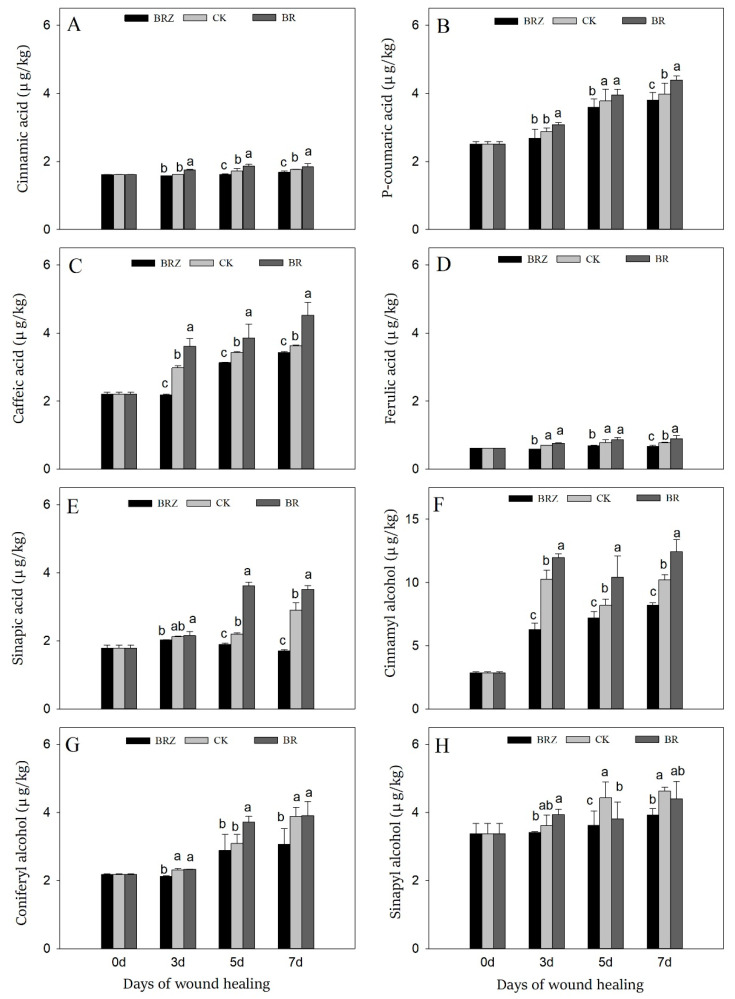
Effects of BR and BRZ treatments on contents of phenolic acids and lignin precursors. Contents of phenolic acids, including cinnamic acid (**A**), p-coumaric acid (**B**), caffeic acid (**C**), ferulic acid (**D**) and sinapic acid (**E**). Contents of lignin precursors, including cinnamyl alcohol (**F**), coniferyl alcohol (**G**) and sinapyl alcohol (**H**). Vertical bars indicate the standard error of three replicate assays. Columns with different letters at each time point are significantly different (LSD, *p* < 0.05).

**Figure 6 foods-11-00906-f006:**
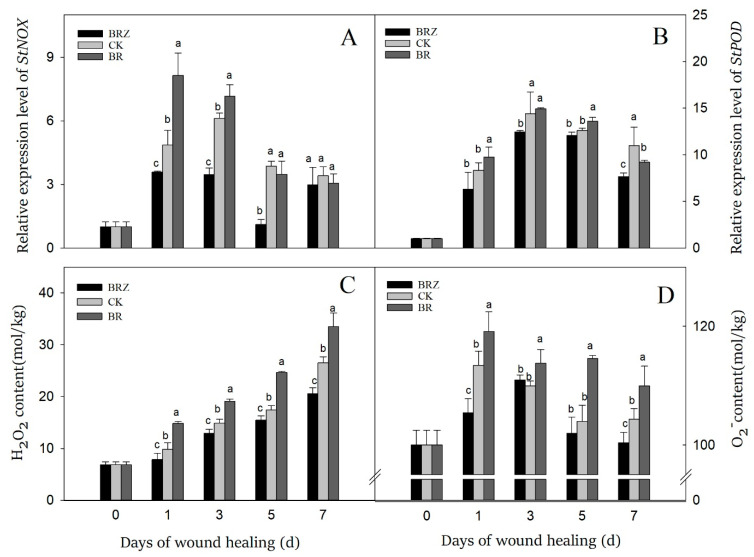
Effects of BR and BRZ treatments on the expression of *StNOX*, *StPOD* genes and the contents of ROS. The expression levels of *StNOX* (**A**) and *StPOD* (**B**) genes. Contents of ROS, including H_2_O_2_ (**C**) and O_2_^−^ (**D**). Vertical bars indicate the standard error of three replicate assays. Columns with different letters at each time point are significantly different (LSD, *p* < 0.05).

**Table 1 foods-11-00906-t001:** Oligonucleotide sequences for primers used in this study.

Gene Name	Accession Number	Primer Sequences (5′–3′)
*StEF* *1α*	PGSC0003DMG400023272	F: ATTGATGCCCCTGGTCACAG R: CATGTTCACGGGTCTGACCA
*StDWF*	PGSC0003DMG400014902	F: AGAGGCGTAATGAAATGA R: TTGAACAGCAGCAGGACA
*StBES*	PGSC0003DMG400027820	F: GTGGGCACAACAACACTAT R: ACACCAGAAAGCCAACCT
*StPAL*	PGSC0003DMG401021549	F: ATGGCTTCTTACTGCTCG R: GGCTACTTGGCTTACGGT
*St4CL*	PGSC0003DMG400014223	F: GTGTTTGCGTTTATTGGC R: GCGTAGTCCTTCACTTTCC
*StCAD*	PGSC0003DMG401025767	F: AAGCTGCTGATTCACTT R: GATGCTCTTTCTCCCTA
*StNOX*	PGSC0003DMG400014168	F: CGGAATCTACTGACATCGG R: CAGCCACAGAGTCTTCACG
*StPOD*	PGSC0003DMG400014055	F: AGGGACTGCTCCATTCTG R: CGGTTATCACCCATCTTA

Letters “F” and “R” indicate the forward and reverse primers, respectively.

## Data Availability

No new data were created or analyzed in this study. Data sharing is not applicable to this article.

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
