# Peer review of "Brassinosteroid Accelerates Wound Healing of Potato Tubers by Activation of Reactive Oxygen Metabolism and Phenylpropanoid Metabolism"

_foods, 2022, doi:10.3390/foods11070906_

Round 1
Reviewer 1 Report
The manuscript entitled "Brassinosteroid accelerates wound healing of potato tubers by activation of reactive oxygen metabolism and phenylpropanoid metabolism" is a nice piece of work by the authors.
Brassinosteroid is an important hormone nowadays increasingly used to alleviate mainly abiotic stress on crops. The increasing number of publications on this topic also shows the importance of studies using this hormone.
Potato grown using BR increased resistance, yield and overall stress resistance. Various scientific strategies are continuously used to reduce the impact of abiotic stressors. Different types of phytohormones are often tested. This paper focuses on investigating the effects of BR on various morphological, physicochemical, yield and quality parameters of Solanum tuberosum L. and on improving wound healing of potato tubers by exogenous application of 24-epicastasterone and brassinazole.
The scientific structure and overall theme of the manuscript are solid and acceptable. BR treatments in the experiment showed positive effects on measured parameters characterizing plant growth, enzyme activity and lignin content. In my opinion, the overall concept is interesting and important. The paper is well written and is a worthy contribution that will be of interest to readers of Foods journal.
The paper requires only minor revisions to make it acceptable for publication, I have a few minor suggestions that I believe will improve the manuscript.
The abstract can serve as a stand-alone document that briefly describes the procedures and conclusions, it is here that I would recommend more specificity. The most important points are specific and present new findings. Overall, the paper is clear and well written.
My suggestions regarding this MS are to accept after revision with the following points:
1. The authors have presented a very good introduction, but they could explain more about the molecular mechanisms of BR action and potential mitigation against abiotic and biotic stress.
2. The authors could have included new aspects in the introduction and discussion - hormonal regulation of tolerance to abiotic stresses (interaction with SA, CYT, etc.), including new references.
3. Add more perspectives on photosynthetic parameters/discuss which types of parameters are more sensitive and why.
4. Add some recent references in MS. It is important to discuss the regulatory mechanisms of plants.
The paper brings many new aspects and the novelty of the paper is fine, but I would encourage the authors to discuss more ecophysiological aspects as well using new references. Add more information about phenylpropanoid pathway, biosynthesis of BR and ROS metabolism; doi: 10.1016/j.scienta.2021.110516; 10.1111/tpj.15416; 10.1105/tpc.112.102574; 10.1007/s11120-020-00708-z; 10.1016/j.foodchem.2020.126875
The manuscript is useful and innovative, containing original data.
This study presents the relevant issues in greater depth than some other related publications, so I recommend MODERATE editing.
Reviewer 2 Report
The manuscript entitled, "Brassinosteroid accelerates wound healing of potato tubers by activation of reactive oxygen metabolism and phenylpropanoid metabolism" is a good piece of work which highlights the functional role of brassinosteroids in potato tubers, in terms of activation of antioxidative responses and phenylpropanoid pathway. The science structure and overall theme of the manuscript are sound and acceptable. In my opinion the overall concept is interesting and important. The paper would be a worthy contribution to the readers of Foods. The paper requires MINOR revisions , I have a few suggestions which I believe will improve the manuscript.
Title should be improved to "Brassinosteroid-mediated wound healing of potato tubers by activation of antioxidative and phenylpropanoid metabolism"
Abstract
The abstract should illuminate the main findings of the paper that can serve as a stand-alone document. However, authors have represented abstract in more generalized form. Authors should emphasize the levels of increment of different parameters assessed in % age values. I suggest revising abstract and incorporate significant results in the abstract for comparative framework of the parameters missing that is antioxidants and oxidative markers. Also, add 1-2 lines describing general overview at the beginning of the MS.
Keywords-add 1-2 more informative words.
Introduction
- The scientific names of plants and pathogens should be italicised in entire manuscript.
- Elaborate their role of BRs in plant growth and development.
- Explain the mechanism of action of BRs mediated defense processes of plants against pathogens.
- Add the adverse effects stresses onto plants in this section in an elaborate manner. Explain properly how they affect plant growth and metabolism. Add a separate paragraph. Then come to BRs and their role.
- Add latest references in the introduction. The recent reports are lacking. Revise the section by adding latest citations.
- Focus on the mechanistic action of BRs towards pathogens and stressors.
- Throw some light on data depicting crop yield losses due to stressors. Latest reports should be cited.
- Line 54-57 Elaborate.
- Also focus on BRs crosstalk with other hormones and molecules during stress management in the introduction.
- Cite the following literature:
Li, S., Zheng, H., Lin, L., Wang, F., & Sui, N. (2021). Roles of brassinosteroids in plant growth and abiotic stress response. Plant Growth Regulation, 93(1), 29-38.
Hafeez, M. B., Zahra, N., Zahra, K., Raza, A., Khan, A., Shaukat, K., & Khan, S. (2021). Brassinosteroids: Molecular and physiological responses in plant growth and abiotic stresses. Plant Stress, 2, 100029.
- Authors should add few lines demonstrating objectives and hypothesis of the research study more clearly.
Material and Methods
Material and methods are well presented.
Results:
Results are well presented.
Discussion
Discussion is more generalized and it is not been explained in the light of the observed results at many places. Please revise the discussion part for such parameters explaining the mechanism of action. I would like to invite authors to discuss more eco-toxicological and eco-physiological aspects using new references.
I found discussion to be very general, more recent work should be cited along with mechanism of action along with observed results. Moreover, I suggest you to re-write the discussion where the recent studies and their findings/conclusions should be discussed appropriately along with their proper explanations.
Read the literature thoroughly and re-write the crux/findings along with the best possible mechanism explained.
Conclusion
Conclusion is well written.
Remove all the errors including spacing and revise the entire manuscript by removing typographical errors.

Reviewer 3 Report
In this study, the authors have determined the role of the brassinosteroid hormone in the wound healing of potato tubers. In the present study, the authors performed mechanical wounding of potato tubers and treated them with BR and BRZ for wound healing. Further, the disease index and weight loss of tubers were analyzed, and gene expression analysis and enzyme activities were determined.
The study topic is interesting and informative, and the overall manuscript is written very well. This manuscript covers overall knowledge on this topic. I suggest authors review the whole manuscript carefully and correct all the mistakes. Authors should improve the grammar, spelling, punctuation, and overall English of the manuscript. The scientific names of the species and the names of the genes must be italicized in the manuscript. The abbreviations should be explained in full during the first mention.
The discussion of the manuscript needs further improvement. In the introduction, discuss the role of BR against biotic and abiotic stresses by citing the following articles.
https://doi.org/10.1007/s10725-020-00672-7; doi: 10.1105/tpc.19.00335.
Before recommending this article for publication, some shortcomings should be resolved.
Page 4-5, lines 75, 76: the references should be cited uniformly.
Page 1, line 15: It should be “stresses”.
Page 1, line 16: It should be “played”.
Page 1, line 30: Scientific names of the species should also be mentioned with common names during the first mention.
Page 4, line 140: Measurement of the contents of…
Page 4, line 152: Measurement of the contents of …
Page 5, line 192: Higher than CK.
Page 5, line 193: Lower than control…
Page 5, line 193-194: “These shown that…did the opposite”. Please rewrite this.
Page 6, line: 207-208: “The gene expression of StDWF and StBES was dramatically …”
Page 11, line 300: Resistance to abiotic and biotic stresses in plants.
Page 11, line 308: Arabidopsis thaliana
Page 11, line 314: StDWF and StBES and enhanced the synthesis…
Page 11, line 314: Which induced tuber wound healing.
Page 11, line 331: Were increased in BR group.
Page 12, line 355: ROS are indispensable…
Page 12, line 356-358: “When tubers damaged…and H2O2”. Please rewrite this.
Page 12, line 359: production of H2O2. Among these, NOX…
Page 12, line 366-367: “However, the increasing effect…was not always”. This sentence is not complete. Please rewrite this.
